# Indoor Navigation in Facilities with Repetitive Structures

**DOI:** 10.3390/s24092876

**Published:** 2024-04-30

**Authors:** Zeev Volkovich, Elena V. Ravve, Renata Avros

**Affiliations:** Braude College of Engineering, Karmiel 2161002, Israel

**Keywords:** indoor positioning systems, repetitive structures, optimal navigation, GPS, Wi-Fi, BLE, iBeacon

## Abstract

Most facilities are structured in a repetitive manner. In this paper, we propose an algorithm and its partial implementation for a cellular guide in such facilities without GPS use. The complete system is based on iBeacons-like components, which operate on BLE technology, and their integration into a navigation application. We assume that the user’s location is determined with sufficient accuracy. Our main goal revolves around leveraging the repetitive structure of the given facility to optimize navigation in terms of storage requirements, energy efficiency in the cellular device, algorithmic complexity, and other aspects. To the best of our knowledge, there is no prior experience in addressing this specific aim. In order to provide high performance in real time, we rely on optimal saving and the use of pre-calculated and stored navigation sub-routes. Our implementation seamlessly integrates iBeacon communications, a pre-defined indoor map, diverse data structures for efficient information storage, and a user interface, all working cohesively under a single supervision. Each module can be considered, developed, and improved independently. The approach is mainly directed to places, such as passenger ships, hotels, colleges, and so on. Because of the fact that there are “replicated” parts on different floors, stored once and used for multiple routes, we reduce the amount of information that must be stored, thus helping to reduce memory usage and as a result, yielding a better running time and energy consumption.

## 1. Introduction

In this paper, we introduce a comprehensive approach and its initial implementation for navigation within repetitive structures such as ships, hospitals, and (partially) underground constructions where GPS signals may not always be available. Our scenario involves a facility with multiple similar components, such as floors. A user (visitor) aims to navigate through the indoor facility without a GPS signal, seeking the optimal route to a desired destination. To address this, we propose the use of a cellular-based application devoid of GPS access but incorporating BLE (Bluetooth low-energy) technology.

Our approach combines iBeacons and cellular hardware devices with navigation software as a unified unit. The complete problem is decomposed into sub-problems, and each stage is discussed independently. We present an efficient algorithmic solution accompanied by relevant illustrations. Lastly, we evaluate the implemented system and discuss the obtained results.

### 1.1. Indoor Navigation

The purpose of this paper is to propose a navigation system that runs on cellular devices, which is not really new. In fact, one of the latest reviews on indoor positioning techniques may be found in [1]. Presently, most of the navigation systems are based on GPS, which has a large deviation problem and is not always accessible. This fact does not allow us to navigate in indoor places and certainly does not allow us to use indoor navigation systems. Furthermore, an additional challenge arises concerning the signal strength of GPS, particularly when utilized in indoor spaces where obstacles such as walls can significantly impact its effectiveness.

Eliminating the reliance on GPS leaves us with Local Positioning System (LPS) and Hybrid Positioning System (HPS). Examples of common wireless communication technologies associated with LPS include ireless Fidelity (Wi-Fi), Bluetooth low-energy (BLE), radio frequency identification (RFID), and ultrawideband (UWB), cf. [2]. Among these options, WiFi and BLE technology emerge as the two primary choices for our specific applications, both being widely utilized for indoor navigation.

Wi-Fi-based indoor localization systems have gained prominence for indoor positioning due to several advantages, cf. [3]. Firstly, most cellular devices are equipped with built-in Wi-Fi modules. Secondly, Wi-Fi access points are widely distributed, making Wi-Fi-based indoor localization systems cost-effective and accessible. Thirdly, Wi-Fi systems do not require additional special-purpose hardware. Recent contributions to this approach can be found in [4] and other related studies, cf. [5].

Bluetooth low-energy (BLE) technology is another viable option for indoor positioning systems. As highlighted in [6], BLE technology stands out for its energy efficiency compared to Wi-Fi, and the deployment of BLE devices is swift due to the easy configuration of BLE tags. BLE indoor localization for building emergency management is presented in [7]; indoor localization for smart heating, ventilation, and air conditioning controls may be found in [8]; indoor localization for point-of-interest identification is described in [9]; and indoor localization for occupancy prediction is given in [10]. The most relevant contributions in our context range from [11,12,13] to [14,15,16,17].

In fact, importance of the original data and their presentation, used in navigation systems, was emphasized in [10]. Here, we take verbatim: *“The first challenge is related to different data sources using inconsistent schemas or data format… Another challenge encountered during POI (Point Of Interest) conflation involves standardising the diverse taxonomies used by different data sources… Lastly, it is also crucial to ensure that the POI matching process is computationally efficient to maintain its viability…”*

This paper proposes a novel framework for performing end-to-end POI conflation involving a six-step approach:Data procurement,Schema standardization,Taxonomy mapping,POI matching,POI unification,Data verification.

The paper seems to be the most relevant to our contribution that we have managed to find as of now.

The approach of [10] was recently used in [18] for establishing a stable received signal strength indication (RSSI) fingerprint value for distance estimation in indoor positioning. It is also mentioned in [19] in the context of vision-based scene recognition.

### 1.2. Novelty of the Proposed Approach

While the aforementioned contributions predominantly focus on location precision, our objective differs significantly. We assume that the user’s location is determined with sufficient accuracy.

Our goal revolves around leveraging the repetitive structure of the given facility to optimize navigation in terms of storage requirements, energy efficiency in the cellular device, algorithmic complexity, and other aspects. We pre-process the original data (maps) to new data structures, which contain some minimal part of the data accompanied by minimal meta-data. This approach allows significant improvement in the data storage requirements. Moreover, all optimal paths are pre-calculated and stored in this optimal way, which significantly improves energy efficiency. It may be considered an optimization sub-step of schema standardization, as provided in [10]. To the best of our knowledge, there is no prior experience in addressing this specific aim.

In our unique application, we substitute GPS satellite signals with iBeacon emitters/sensors (refer to Figure 1) due to their energy efficiency and ease of deployment. To achieve our objective, real-time user location information is crucial, obtained through BLE- based devices, specifically iBeacons. As the user moves, there is a necessity to receive and process data in real time from the iBeacons.

Our implementation seamlessly integrates iBeacon communications, a pre-defined indoor map, diverse data structures for efficient information storage, and a user interface (UI), all working cohesively under a single supervision. Each module can be considered independently, followed by an integration process encompassing code combination in various programming languages and handling different types of hardware devices.

Our primary accomplishment is the successful navigation of the users from their current location to the chosen destination. The application presents navigation instructions, the current location, and the route on the screen, along with loaded route information. Additionally, in the event of a navigation mistake caused by the detection of an unexpected iBeacon on the route, the application loads and updates the corresponding changes to redirect the user.

Furthermore, the application is designed with optimal time and space complexity, considering hardware constraints such as relatively small memory and computational capability. The minimum requirements for our system include 2 GB of RAM (sufficient for Android 10 OS), Bluetooth 4.0 for BLE communication support, and a quad-core CPU up to 1.3 GHz. Given these constraints, our application relies on pre-calculated sub-routes, scans iBeacons in low-power mode for battery conservation, and implements reused routes. As part of defining the success criteria, we can maintain minimal computing time and memory usage.

Additionally, the application architecture incorporates several threads for iBeacon communication, UI updates, and a route thread. These main threads necessitate the implementation of synchronization methods to ensure efficient and reliable application activity.

### 1.3. Implementation Challenges and Solutions

In this section, we outline several implementation challenges encountered and the corresponding solutions:

**Description:** Non-optimal iBeacon scanning.**Difficulty:** The BLE system callbacks for iBeacons were too fast to detect the iBeacons’ PDU (protocol data unit) compared to their emitters’ broadcast. For this reason, the application received incomplete portions of information.**Solution:** We decelerated the BLE callbacks by the system low-power mode BLE callbacks to fit the iBeacon broadcast emitters. Moreover, we customized the iBeacon broadcast emitter to 400 ms. After the changes, the system stabilized and the iBeacons discovery times were improved significantly by an average of 150–200 ms instead of 380–450 ms.**Description:** Synchronization problem between a cellular device and an iBeacon.**Difficulty:** In the iBeacon scan process, when the cellular device is physically moved, the signals from the iBeacon were not received nearly in real time. Additionally, we observed that the callbacks for the BLE scan on a cellular device commenced approximately every 40 ms to 90 ms on average, while the iBeacon emitter broadcasted its data packet every 900 ms.**Solution:** We downloaded an “AprilBeacon” application for programmers and changed the broadcast frequency from every 900 ms to 400 ms to receive synchronized time between the iBeacon emitter and the cellular device BLE callbacks.**Description:** One-to-one iBeacon ID number, major and minor variables.**Difficulty:** We should use the major and minor variables of iBeacon’s IDs in the system to attribute them to iBeacons in our application.**Solution:** We installed and used an in-shelf product from the manufacturer to program each iBeacon with major and minor properties to classify them into junctions inside routes or facilities in the application.**Description:** Presenting routes to the destination and the current location on UI.**Difficulty:** The real-time presentation of the user location by the nearest perceived iBeacon posed a challenge. The main difficulty involved determining a suitable method to share the current iBeacon and the corresponding route to the destination in real time between the main UI thread and the iBeacon scanner thread.**Solution:** Dealing with the difficulties, we built a bitmap to present and update in real time the current location and the route coordinates on the specific UI screen canvas by shared memory variables that specify the route and the current iBeacon.**Description:** Building and storing the pre-calculated sub-routes in an optimal way.**Difficulty:** Memory constraints occurred in cellular devices.**Solution:** We constructed hash maps and shared the same route objects across multiple data structures to optimize memory usage and enhance application efficiency.**Description:** Data sharing between an iBeacon scanning thread and a UI thread.**Difficulty:** Eliminating dependencies between threads, implementing synchronization methods such as mutexes to protect critical sections, and maintaining a handlers’ queue to update the main UI thread accordingly. Managing system threads in real-time iBeacon scanning and UI updating is crucial, and data sharing between threads necessitates synchronization.**Solution:** We employed synchronization methods to eliminate dependencies and address the producer (iBeacon scanning thread)–consumer (UI thread) issue for an appropriate route. Additionally, we constructed a handler queue for the main UI thread, containing handlers to publish on the UI view in the main UI thread fragment.

### 1.4. Structure of the Paper

The paper follows the subsequent structure: In Section 2, a review of current navigation technologies is presented. Section 3 provides a technical overview of the utilized technology. Our approach to positioning iBeacons in repetitive structures is introduced in Section 4, while Section 5 outlines the data model. The primary content of the paper is found in Section 6, which details the proposed navigation algorithm and includes examples of its application. Finally, Section 7 concludes the paper, offering a summary of the results and future outlook.

## 2. General Background

Here, we review technologies currently employed in different cases of navigation.

### 2.1. Global Navigation Satellite Systems

A satellite navigation system (GNSS—Global Navigation Satellite System) is a system of ground and space equipment designed for positioning in space and time, as well as for determining the speed, direction, and other parameters of the movement of an object. Common elements of a satellite navigation system:

**Orbital group**—System of spacecraft in the form of a network of navigation satellites;**Ground command and control system**—Blocks for measuring the position of satellites and transmitting the received information to them to correct information about orbits;**Reception equipment—**“Satellite navigators” used to determine location;**Optional information radio system**—for transmitting corrections to users, which can significantly increase the accuracy of coordinate determination.

The principle of operation of satellite navigation systems is based on measuring the distance from the receiver antenna at the site to navigation satellites, the location of which is known with great accuracy. The method of measuring the distance from a satellite to a receiver antenna is based on determining the speed of propagation of radio waves.

The most famous satellite navigation systems today are GPS (Global Positioning System), GLONASS, and Galileo. They all work on a similar principle: for average positioning accuracy in space, the receiver antenna must receive a signal from at least four satellites of the system (or from three, if one of the coordinates is known: for example, the altitude of a ship on the ocean is 0 m). However, there are certain differences. For example, each satellite navigation system determines a location in its “own” coordinate system; each satellite navigation system belongs to a different country or group of countries.

### 2.2. Mobile Positioning Systems

All technologies related to determining location in cellular networks are called Mobile Location Services (MLS). In this case, one should distinguish between services for determining the exact position, such as x, y coordinates or location-based service, and services tied to the user’s location (district, region, location-dependent service). In the first case, these are navigation services, obtaining up-to-date information related to coordinates. In the second one, knowledge of exact coordinates is not required; the system operates with the concept of location.

### 2.3. Wi-Fi Positioning System (WPS)

The Wi-Fi positioning system (WPS) has been a subject of study in various fields, gaining increased attention from mobile companies. This paper introduces an indoor Wi-Fi positioning system designed for Android-based cellular devices. WPS typically utilizes Wi-Fi signals from existing private and public Wi-Fi access points (APs) to deliver location-based services (LBS). It serves as a valuable complement to the global positioning system (GPS) in urban centers and indoor environments. Previous studies of WPS have explored different approaches, such as utilizing Wi-Fi signal strength for position calculation, employing a Kalman filter (KF) for signal stabilization, and integrating Wi-Fi with GPS for improved accuracy. However, Wi-Fi signals may still provide lower precision for location tracking.

### 2.4. Bluetooth

Bluetooth, a radio technology, facilitates short-distance communication and enables Bluetooth positioning by measuring radio wave signal intensity values. This wireless technology operates globally within the 2.4 GHz band, which is freely available for use. Utilizing Bluetooth technology does not incur additional costs beyond the purchase of the desired Bluetooth device. One notable advantage of Bluetooth indoor positioning is the small size and low power consumption of Bluetooth chips, making them easy to integrate into mobile phones and other compact devices. However, drawbacks include higher costs and potential instability in complex spatial environments.

### 2.5. BLE and iBeacons

iBeacon is a device with Bluetooth low-energy (BLE) technology, which, when a client approaches with an Apple iPhone (iOS) or Android, displays a standard notification on the smartphone or launches a mobile application. The iBeacon constantly emits a Bluetooth radio signal, and smartphones detect it at a distance of up to 50 m and accurately determine its location. In simple words, it can be described as a more accurate GPS that works indoors. iBeacon technology, for example, allows you to park your car in an underground garage and then pinpoint the path to find it after you have completed your shopping. And when you go to a large supermarket, iBeacon will be able to pave the way for you to the shelf with the product you are interested in.

## 3. Technical Overview of the Used Technology

In this section, we review the iBeacon’s specification and the location accuracy achievable with a limited number of iBeacons that are placed inside a room. The iBeacon’s location accuracy is indicated by the received signal strength indicator (RSSI) levels that the iBeacon emitters transmit. For the specification of the “AprilBrother” iBeacons used in the project, see Figure 1.

In general, one may use iBeacons from other suppliers with the corresponding technical tuning of a particular implementation.

iBeacons’ communication frequency consumption open to the public is between 2400 and 2483.5 MHz. The received signal strength indicator (RSSI) may be used in order to determine the iBeacon emitter’s distance to the cellular device. The distance measures are divided into three areas: close (d≤2 m), near (2 m ≤d≤ 5 m), and far (5 m ≤d≤ 26 m).

An iBeacon has BLE proximity sensing technology. It can transfer a uniform code unique ID (UUID) to an application’s intelligent terminal. The obtained information of UUID and RSSI may be converted into more iBeacon features.

### 3.1. BLE Data Packet

The BLE data packet is composed of four parts, as explicitly shown in Figure 2. The most important block for us is the PDU (protocol data unit) data. In Figure 3, we show an example of this block split into sub-blocks, taken verbatim from [21]:

**The proximity UUID:** (B9…6D in our example) is an identifier which should be used to distinguish your beacons from others. If, for example, beacons were used to present special offers to customers in a chain of stores, all beacons belonging to the chain would have the same proximity UUID. The dedicated iPhone application for that chain would scan in the background for beacons with the given UUID.**The major number:** (2 bytes, here: 0 × 0049, so 73) is used to group a related set of beacons. For example, all beacons in a store will have the same major number. That way the application will know in which specific store the customer is.**The minor number:** (again 2 bytes, here: 0 × 000 A, so 10) is used to identify individual beacons. Each beacon in a store will have a different minor number, so that you know where the customer is exactly.**TX power:** TX power, known as measured power, is a factory-calibrated, read- only constant that indicates the expected RSSI at a distance of 1 m to the iBeacon. Combined with RSSI, it allows estimation of the distance between the device and the iBeacon.

### 3.2. Specification of the Used iBeacon Emmiter

Specification of the used emitter is provided on Figure 4. The main useful characteristics are:May be used standalone as an iBeacon;External host is not required;Allows application for advertisement and location;Built-in pairing password to prevent others from modifying the settings;Supports customization of your own iBeacon configuration, including UUID, etc.;TX power is configurable;Advertising frequency is configurable, etc.

**Figure 4 sensors-24-02876-f004:**
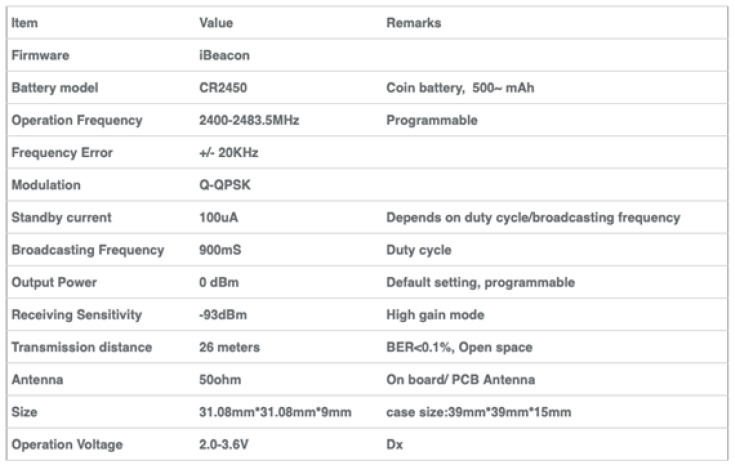
Specification of the used iBeacon emmiter, cf. [20].

## 4. iBeacons’ Positioning

Let us take a huge cruise liner with stairs and elevators as an example of a facility with a repetitive structure. We suggest initially situating all iBeacons around attractions or other points of interest, taking the navigation within a ship, as illustrated in Figure 5.

This concept can be readily adapted to other comparable repetitive facilities. Indeed, when navigating within a ship, such as reaching a restaurant, one traverses corridors and utilizes elevators and stairs, encountering various points of interest (POIs). The ship’s structural layout enables the strategic placement of iBeacons along corridors, staircases, and elevators to achieve comprehensive 3D navigation coverage. We designate these POIs as nodes.

The placement strategy must adhere to crucial positioning criteria, including localization coverage, success, and accuracy. Let Li represent the location of Nr, where BNr is the identifier of an iBeacon associated with a specific POI. These Nr values must be chosen in a manner that ensures:every position (for users’ cellular devices) must be covered by at least one iBeacon (localization coverage);every $L_{i}$ of every $BN_{r}$ must be non-collinear (localization success and accuracy);the shortest possible distance between $L_{i}$ and the cellular devices must be $BN_{r}$ reachable (localization precision);an optimal number of iBeacons must be deployed for maximum 3D navigation coverage (efficient network cost).

In addition, we must take into account that (in the indoor layout) all users must be serviced from the height of a wheelchair (60 cm) to an average person as tall as 175 cm.

## 5. Data Model Overview

In this section, we outline our data model and its implementation in our algorithm.

Currently, navigation applications like Google Maps employ extensive graphs with nodes and edges to determine optimal travel routes. Utilizing real-time algorithms, they calculate the fastest or shortest paths from one point to another. In contrast, our approach aims to simplify calculations by relying solely on a pre-calculated database (DB) of routes and their combinations, as detailed in this section.

Moreover, conventional navigation applications often rely on cloud-based services for storing data and additional information. In our specific application scenario, however, we assume a lack of access to internet services, precluding the use of cloud-based storage for our database (DB). Consequently, our only option is to utilize the local storage capacity of the cellular device to store the routes in the DB. This constraint necessitates an extremely compact data presentation.

The DB model is constructed around two main structures, both represented as vectors. The primary vector, denoted as CR (Complete Route), consists of groups of secondary vectors, denoted as SR (Sub-Route). Combinations of these vectors form a unified data structure. This design ensures time-optimal navigation without incurring unnecessary battery drain, representing a key advantage of the proposed DB structure.

Now, we describe the data structure from bottom to top. The top data structure refers to a route and the bottom data structure refers to the sub-routes, as illustrated in Figure 6.

### 5.1. Node Allocation Table (NAT)

As it was shown above in Section 3.1, for BLEs, the most important block for us is the PDU data; see Figure 3. We use the major number to encode the floor (deck) number and the minor number to encode a particular facility on the floor; see Figure 7.

In fact, some facilities may be unique, like restaurants or cinema halls. In these cases, without loss of generality, first, one should reach the corresponding floor and then, walk on this floor from the elevator (or stairs) to the desired facility.

Otherwise, if we have the “same” facilities on different floors, then all of them have the same minor number. In fact, the same approach is mostly used for numbering of rooms in hotels. As a rule, the first two digits present the number of the floor, while two or more digits are for the room number on the floor. Let us take two rooms, say, 1023 and 1123, which correspond to the “same” rooms (number 23) on floor 10 and on floor 11, respectively, placed one above the other. On the other hand, in this case, the major number, which corresponds to the floor, may be used as well in navigation. In fact, if you are going to room number **10**23 then you need to first go to floor number **10** by elevator or stairs.

Thus, we split the general case into two sub-cases, accordingly.

**First case:** A unique NAT is assigned for each facility that contains the entire iBeacon group up to the desired destination. This NAT, as we explained, will be a single one and there is no additional copy of it that is relevant on other floors (since it is located on a specific floor).**Second case:** A facility is located on a floor that contains, say, living rooms, which probably contain additional iBeacons ($NAT_{i}$).

A NAT (Node Allocation Table), see Figure 8 (NATs are marked by red rectangles), is a vector of 1D elements: minor numbers of BLEs. Each particular NAT incorporates all the iBeacons that are in a common area. There may be the same $NAT_{i}$ in different decks, in cases where the decks’ structures are identical.

The characteristic of that area is that it has no junction/elevator/stairs and, therefore, in the macro analysis of the route calculation problem, all the iBeacons can be treated as one unit by the proposed algorithm. Note that for each NAT, a dedicated variable will be maintained by the algorithm, and its purpose is to guide the user in which direction within the common area she/he should move.

### 5.2. Junction (JNC) Presentation

When the user reaches a point on the map where it is possible to move in more than one direction (right/left, up/down) and a pre-defined iBeacon is assigned to the point, then the point will be defined as a junction (JNC). The purpose of the junction is to connect different NATs from the same floor or on different floors by stairs/ elevators.

For example, a user wants to go from $NAT_{1}$ on the second floor to $NAT_{1}$ on the third floor. The user will move along the route until reaching a suitable junction (elevator/stairs), where the application directs them to go up one floor. From that point, the user reaches the third floor and continues according to the route to $NAT_{1}$ on the third floor.

A junction in a data structure is characterized by JNCX.N.D, where

X is the floor number;

N is the junction number in floor X;

D is the direction, as shown on Figure 9.

Examples of such junctions are rather in Figure 10.

### 5.3. Complete Route Presentation: 2D Case

The complete routes are presented by vectors, which encode the route from the user’s origin to their destination. The vector consists of NATs and JNCs, which describe the composition of the sub-route towards the target. Moreover, after any $NAT_{i}$ on the route is reached, we add a match variable at the following place in the vector that presents the route direction. Each element in the vector is a part of the complete route. The vector structure is as follows:$$CR<ble_{origin}><ble_{destination}>=\left[NAT_{i},JNC_{i},...,JNC_{k},NAT_{k}\right]$$
For example, see Figure 11:$$CR<1><9>=[NAT_{1},JNC1.1.5\left(Straight\;forward\right),NAT_{3},JNC1.2.4\left(Left\right),NAT_{5}],$$
where

<1>: origin iBeacon;<9>: destination beacon;JNC1.1.5 (5): at Junction JNC1.1 on floor 1, keep moving straight forward; see Figure 9;JNC1.2.4 (4): at junction JNC1.2 on floor 1, turn left; see Figure 9.

**Figure 11 sensors-24-02876-f011:**
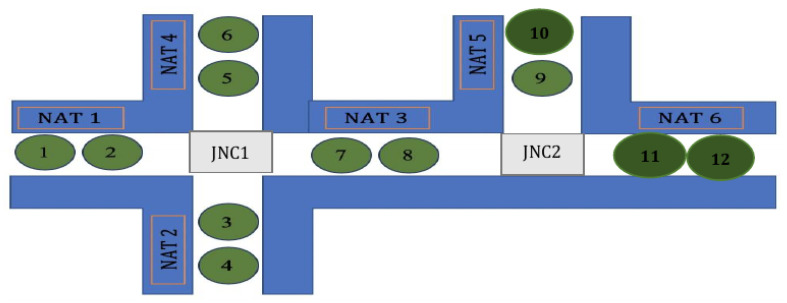
iBeacons, their numbers, NATs, and JNCs: a 2D example.

The main idea is to pre-calculate some CRs, store them in an optimal view, and use them in construction of more complicated routes. In our implementation, we use a matrix that contains references to all pre-calculated CRs. The references point to a data structure (as in the provided example) that encodes the pre-calculated CR.

Our implementation is based on hash maps. Each element in the hash map contains two values: a string that represents the requested route from origin BLE to destination BLE (the key) and a vector (array) that represents the corresponding CR (the value).

### 5.4. Complete Route Presentation: 3D Case

We start from the example shown on Figure 11 for one deck. The circles with numbers are the BLE devices, where each number is a BLE identifier (minor number). Now, we want to extend it to the case of two or more similar floors (decks). Our model expands the corresponding version of the previous model of one floor to 3D navigation. This model uses a similar data structure as in the 2D model. However, in this case, we should assign $JNC_{i}$s with up/down instructions; see Figure 12.

The architectural layout of each floor on the ship is predominantly identical, as illustrated in Figure 12. This implies that BLE devices will be positioned in the same locations. However, in certain instances, there might be floors that deviate from the identical layout, resulting in variations in the positions of BLE devices.

Here, we provide an example with a data structure that corresponds to the case when a user starts navigation from an origin numbered BLE 1 on the first floor and goes to the destination on the second floor (2) numbered BLE 12. The user will be directed from one BLE device to another one on the same floor and then (by using an elevator or stairs) will be directed to another floor.

In our specific scenario, BLE numbering comprises two digits. The first digit indicates the major number, representing the floor number, while the second digit signifies the minor number, indicating the room number on the floor. This numbering system is also used for identifying the user’s current position; see Figure 13.

In instances where we have the “same" facilities on different floors, all of them share the same minor number. For example, the trip might be from the first location (BLE number 1 on $NAT_{1}$ floor 1) to another location (BLE number 2 on $NAT_{6}$ floor 2). We manage the navigation through these locations on different floors.


**Navigation instructions:**
The user will be guided to go through $NAT_{1}$ in the ascending order of the iBeacons’ numbering until they reach the $JNC_{1}$ junction.According to JNC1.1, the user will be guided to go straight forward to $NAT_{3}$.The user will go through $NAT_{3}$ in ascending order of the iBeacons’ numbering until they reach the JNC1.2 junction.According to JNC1.2 on the first floor, the user will be guided to go up (elevator/stairs).According to JNC2.2 on the second floor, the user will be guided to go right to $NAT_{6}$.The user will go through $NAT_{6}$ in ascending order of the iBeacons’ numbering until they reach the destination.


The corresponding CR data structure will contain the following data:$$CR=[NAT_{1},JNC1.1.5\left(Straight\;forward\right),NAT_{3},JNC1.2.2\left(Up\right),JNC2.2\left(Left\right),NAT_{6}].$$

Each element in the CR corresponds either to a junction iBeacon or to a set of iBeacons: a $NAT_{i}$. Moreover, $NAT_{1}$, $NAT_{3}$, and $NAT_{6}$ are the same for each such deck according to the used numbering and locating strategy. The last allows us to store each one of them only once and reduce the used storage space.

## 6. 3D Navigation in Facilities with Repetitive Structures

In our scenario, BLE devices are strategically placed throughout specific areas, like a cruise ship, as depicted in Figure 5. A passenger’s cellular device picks up Bluetooth signals transmitted by these BLE devices, enabling accurate turn-by-turn navigation. This navigation system guides passengers from their current location to any desired point of interest (POI).

### 6.1. The Main Algorithm

Our algorithm proceeds as follows; see Figure 14:

**Input:** Current location, translated to an iBeacon number; desired destination, translated to another iBeacon number.**Output:** Navigation directions and instructions to destination iBeacon.

**Figure 14 sensors-24-02876-f014:**
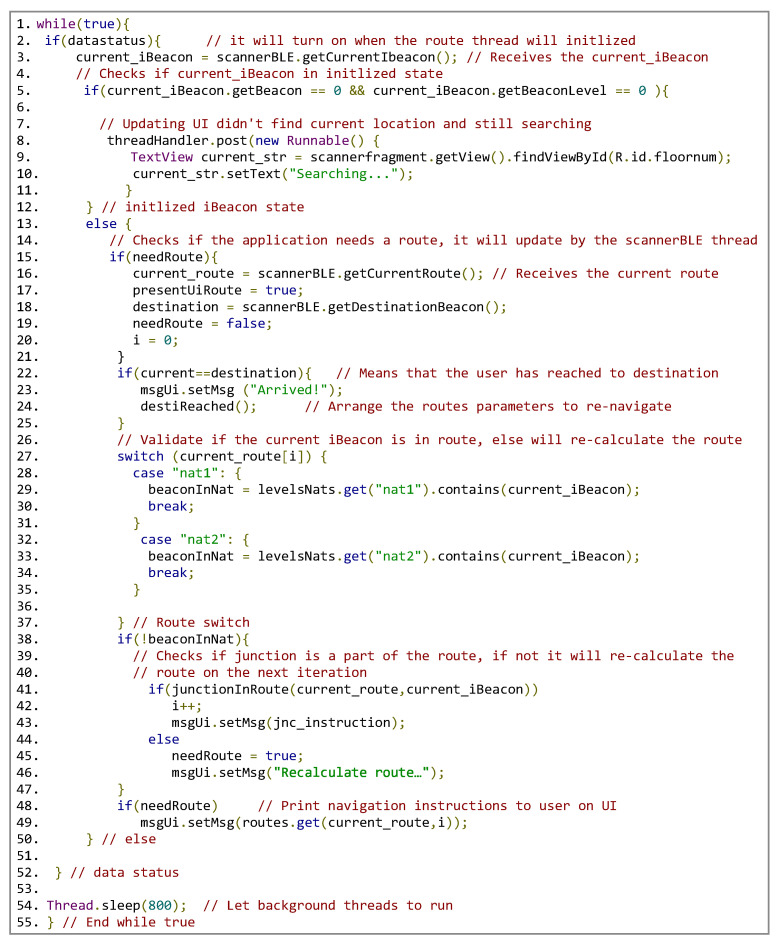
The main algorithm: pseudo-code.

### 6.2. Using the Main Algorithm: An Example

To illustrate our implementation, we created a digital map of a building on our campus, specifically the EM building. In this example, we demonstrate navigation from iBeacon number 111 (located on floor 1 at $NAT_{1}$, corresponding to class EM200) to another iBeacon, number 121 (on floor 1 at $NAT_{2}$, corresponding to class EM203).

The subsequent example outlines the system’s execution in the route thread, when the junction iBeacon, numbered 191 (100—floor number 1, 91 is the junction id number in floor 1), is detected by the scannerBLE thread. For clarity, we refer to a specific CR: $[NAT_{1},JNC1.1.5\left(Left\right),NAT_{2}].$

The subsequent example is depicted in Figure 15 and Figure 16.

The corresponding login screen and the navigation map of a floor in building EM are shown in Figure 17. The displayed route is shown in Figure 18.

## 7. Conclusions and Outlook

The expected results were:The use of the repetitive structure of the floorplan for different floors (by using NATs) to help us make an easy transition between different floors.Success in performing the proof of concept (POC) of navigation when the user is navigating according to the directives of our application. Otherwise, when navigating with errors, the error must be discovered and the route should be re-calculated. Then the navigation via the new route should be continued.Performing navigation that starts from one location point by using the nearby iBeacon.

The reality is as follows:The user is able to navigate with the application after ’=the “GO" button is pressed:(a)Detection of the current location is carried out;(b)The system verifies that the destination was selected.When the user gets close to an iBeacon (by a higher value than a constant RSSI value of 3–4 m), the system detects the iBeacon object in the data structures and marks this place as the current location of the user on the map with a red circle.The system displays the route as a red line according to the user’s choice.The system displays instructions and plays recorded audio files appropriately to the routes in/out NATs through junctions among floors.When navigating with errors, the error is discovered and the route is re-calculated.The system notifies the user when the destination has been reached.The system allows navigation again after reaching the destination.

The proposed and implemented data structures meet the minimal storage requirements, allowing the real-time operation of the application on the limited capabilities of cellular devices.

However, there is room for significant improvement in the navigation algorithm, particularly in optimizing the expected load on the selected routes. Typically, users visit restaurants and other facilities such as cinemas almost simultaneously. Therefore, it is crucial to direct them via different routes as much as possible to prevent over-loaded pathways.

Moreover, we (as an academic institution) do not have enough ability to provide results from testing in different relevant environments, such as passenger ships, hotels, or colleges, to demonstrate the algorithm’s effectiveness. We rather provide a proof of concept and leave a comprehensive evaluation and testing of the complete system to our colleagues from industry.

## Figures and Tables

**Figure 1 sensors-24-02876-f001:**
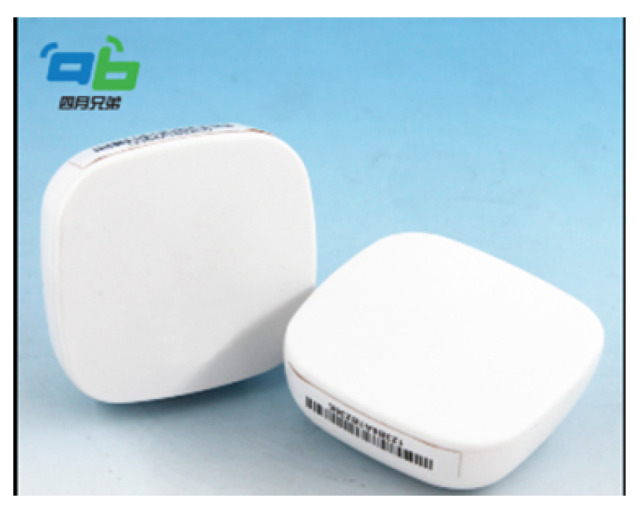
An iBeacon emmiter, cf. [20].

**Figure 2 sensors-24-02876-f002:**
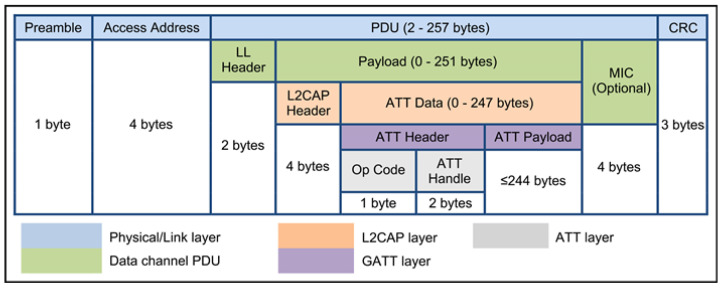
BLE data packet, cf. [22].

**Figure 3 sensors-24-02876-f003:**
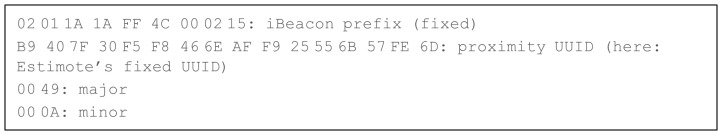
An example of a BLE PDU sub-packet, cf. [21].

**Figure 5 sensors-24-02876-f005:**
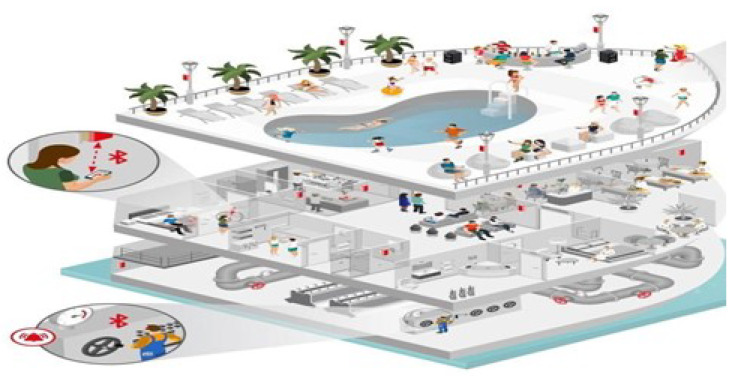
Navigation in a ship, cf. [23].

**Figure 6 sensors-24-02876-f006:**
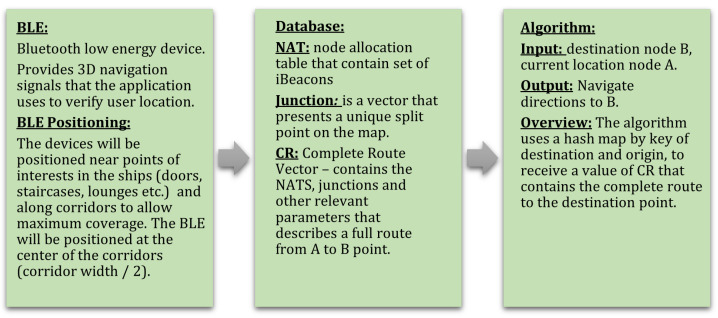
Data model and algorithm.

**Figure 7 sensors-24-02876-f007:**
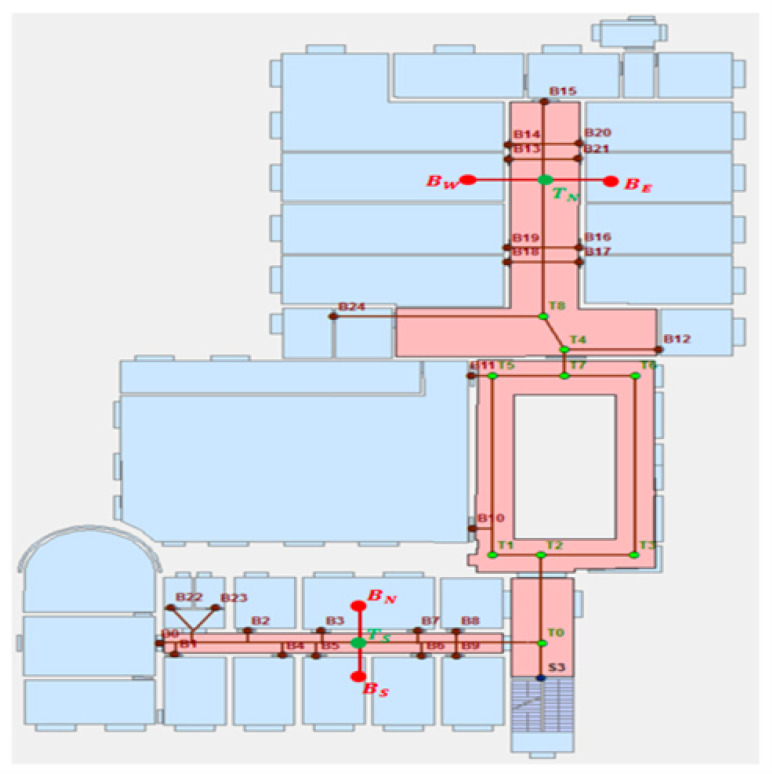
BLE positioning and numbering: an example.

**Figure 8 sensors-24-02876-f008:**
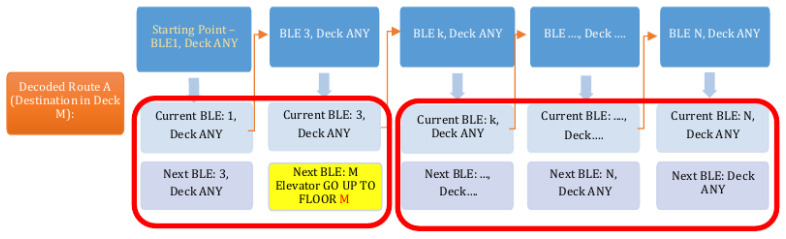
NAT numbering and use in a route presentation: an example.

**Figure 9 sensors-24-02876-f009:**
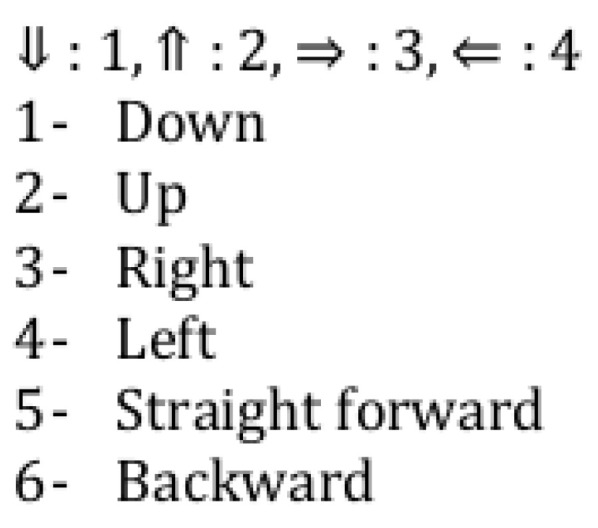
Possible directions.

**Figure 10 sensors-24-02876-f010:**
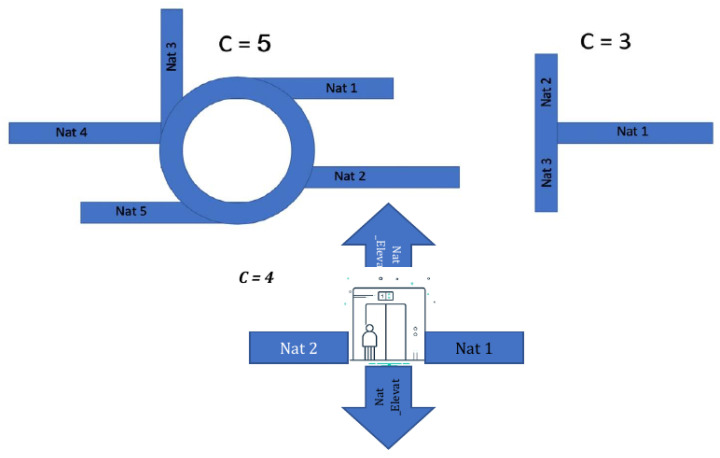
Examples of different junctions.

**Figure 12 sensors-24-02876-f012:**
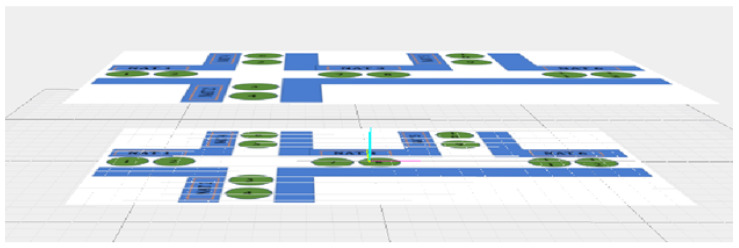
iBeacons, their numbers, NATs, and JNCs: a 3D example.

**Figure 13 sensors-24-02876-f013:**
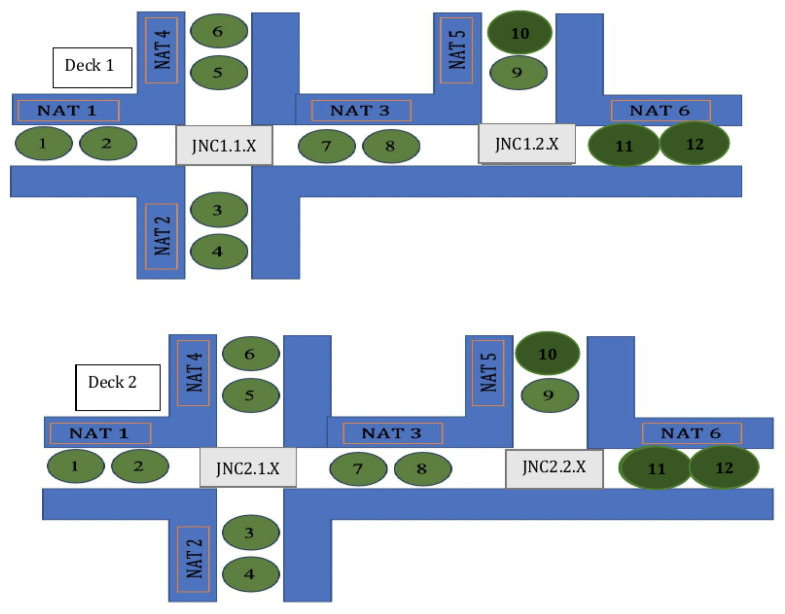
iBeacons located on different decks: an example.

**Figure 15 sensors-24-02876-f015:**
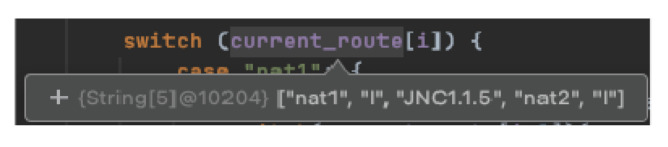
The current route expression in the navigation application.

**Figure 16 sensors-24-02876-f016:**
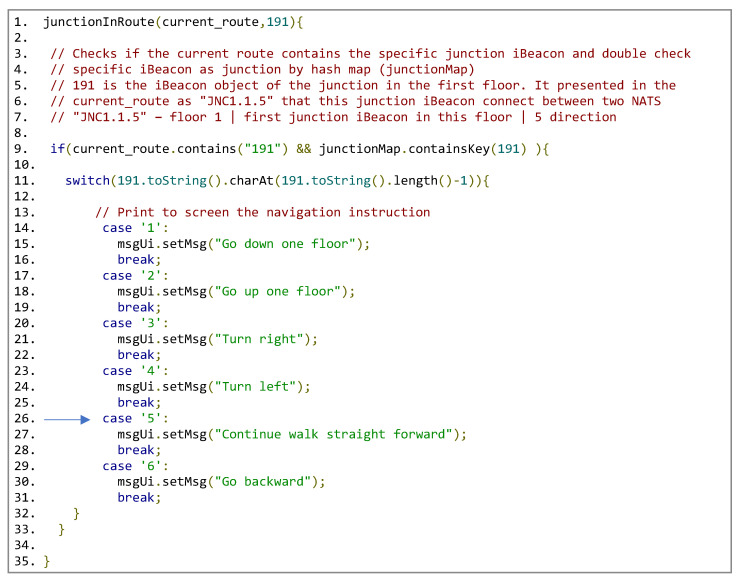
Run of junctionInRoute(current_route,191).

**Figure 17 sensors-24-02876-f017:**
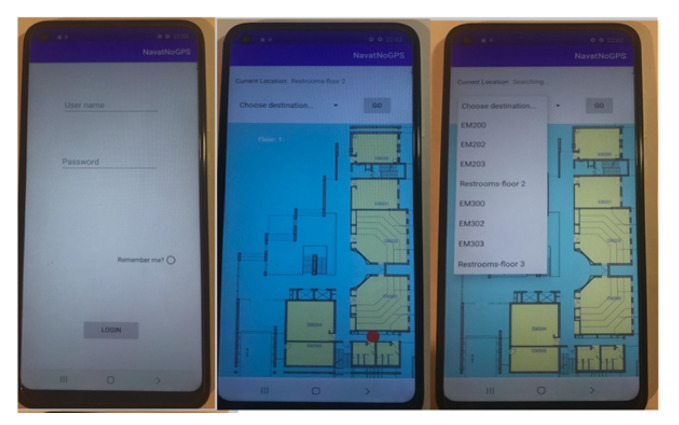
The login screen and the navigation map of a floor.

**Figure 18 sensors-24-02876-f018:**
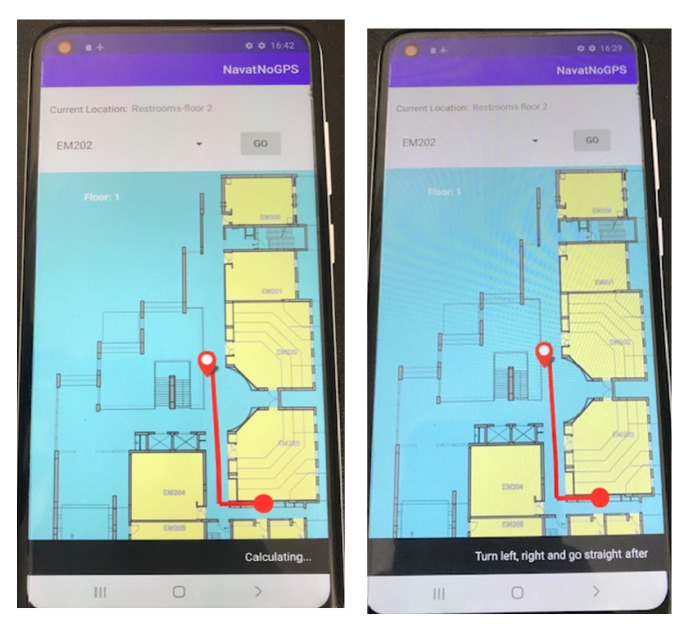
The displayed route.

## Data Availability

Data are contained within the article.

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
