# Peer review of "Indoor Navigation in Facilities with Repetitive Structures"

_sensors, 2024, doi:10.3390/s24092876_

Round 1

Reviewer 1 Report

Comments and Suggestions for Authors
  1. 1. The proposed work does not explicitly state the contribution or novelty of the algorithm. Clearly articulating how this work advances the current state of the field is essential for acceptance.

  2. 2. The mention of providing high performance in real-time raises questions about the extent of real-world testing and validation. The paper should include results from testing in relevant environments, such as passenger ships, hotels, or colleges, to demonstrate the algorithm's effectiveness.

  3. 3. The abstract indicates a partial implementation of the algorithm. A comprehensive evaluation and testing of the complete system is necessary for a successful submission.

  4. 4. Most of the figures are unclear, and in Figure 4, you have presented a table as an image.

  5. 5. General representation of images, code, and tables is not up to the standard

Comments on the Quality of English Language

Moderate editing of English language required

Reviewer 2 Report

Comments and Suggestions for Authors

Comments to the Author

This paper presents an algorithm and its partial implementation for a cellular guide in facilities without the use of GPS. However, there are several points that need to be addressed to improve the quality of the manuscript.

Suggestions to improve the quality of the paper are provided below:

1.     The structure of the manuscript does not conform to a regular academic paper with a proper introduction covering the background and context of the problem, a literature review describing the current state-of-art, and sufficient references to substantiate claims made in the paper. Please kindly review the structure of past literature that has been published and restructure the paper accordingly.

2.     Furthermore, the introduction section should also describe some of the potential applications of indoor localisation. Some of the popular applications include building energy management, emergency management, point-of-interest identification, and occupancy prediction. Please kindly review these papers as a good starting point and highlight some additional applications of indoor localisation as much as possible:

BLE Indoor localisation for building emergency management

10.1109/IUCC-CSS.2016.013

BLE Indoor localisation for smart energy management

https://doi.org/10.1016/j.buildenv.2022.109472

Indoor localisation for smart HVAC controls

https://doi.org/10.1145/2517351.2517370

Indoor localisation for point-of-interest identification

https://doi.org/10.3390/ijgi10110779

Indoor localisation for occupancy prediction

https://doi.org/10.1016/j.buildenv.2022.109689

3.     The authors should combine Section 1.2 and Section 1.4, and focus on describing how the proposed approach extends upon existing indoor localisation approaches instead of listing out the proposed system features.

4.     The General Background section (equivalent to literature review) provides a high-level overview of the various technologies for indoor navigation but does not reference any past studies nor provides a critical of those studies. Please review specific studies that are similar to this study and provide a critical review of the limitations of those studies.

5.     While the technical complexity of this manuscript is sufficient, the manuscript appears to be incomplete as it does not include a case study to demonstrate the performance of the proposed indoor localisation system. Therefore, it is difficult to determine the validity of the proposed approach.

6.     Minor Comments

·      I would strongly suggest against directly using images from other sources in this manuscript (e.g., Figure 3 and Figure 5) to avoid any copyright issues.

·      Please replace Figure 14 with pseudo code instead of the actual code implementation.

Comments on the Quality of English Language

There are some moderate issues related to the manuscript's quality of English, with the main issues highlighted in my current set of comments.

Round 2

Reviewer 1 Report

Comments and Suggestions for Authors

The reviewer agrees with the modifications. 

Author Response

Dear reviewer,

we have not found any remark.

Thank you very much for your valuable advises,

Elena.

Reviewer 2 Report

Comments and Suggestions for Authors

Thank you for taking the time to update the manuscript based on the comments provided. However, there are still two main comments that were not addressed in the latest version of the manuscript.

1. The General Background section (equivalent to literature review) provides a high-level overview of the various technologies for indoor navigation but does not reference any past studies nor provide a critical of those studies. Please review specific studies that are similar to this study and provide a critical review of their limitations.

I still did not find any references to relevant studies on indoor navigation in the General Background section. It does not have to only consist of papers that use a repetitive structure. Other relevant studies on indoor navigation should be included to highlight the novelty of this work.

2. While the technical complexity of this manuscript is sufficient, the manuscript appears to be incomplete as it does not include a case study to demonstrate the performance of the proposed indoor localisation system. Therefore, it is difficult to determine the validity of the proposed approach.

I believe the authors can still perform a very simple experiment within a small study area (such as the authors' office area) to properly test out the POC and evaluate its performance under different scenarios. When testing out the POC, it is also important to set proper evaluation metrics to objectively evaluate the proposed approach. For instance, what is the localisation accuracy of the approach? If reducing storage requirements is one of the features of the approach, what are the storage requirements before and after the system's enhancements, etc? All of these need to be properly laid out in a Results section.

Comments on the Quality of English Language

There are no major issues related to the manuscript's quality of English, except for some minor issues that do not affect the clarity and flow of the manuscript.

Author Response

Dear reviewer,

about the relevant work: we looked again for papers in the same direction, and we have not managed to find any similar attempt. Most investigation are dedicated to precision issues. We expanded a previously mentioned paper [10] and explained how it is connected to our contribution.

About more experiments: as you may be known, we are in war now and just not able to conduct any additional experiment in at least one year. As we emphasized, the main contribution of our research is the use of a new data structure, which contains both data and meta-data that for sure minimizes the memory requirements and pre-calculated optimal route, which for sure minimizes the energy requirements.

Many thanks for your valuable advises,

Elena.

Round 3

Reviewer 2 Report

Comments and Suggestions for Authors

Thank you for taking the time to address my comments thoroughly and comprehensively. I believe all my comments have been adequately addressed, and the quality of the manuscript has increased significantly as a result. I have determined that the manuscript is now ready for publication.

Comments on the Quality of English Language

There are no major issues related to the manuscript's quality of English, except for some minor issues that do not affect the clarity and flow of the manuscript.